# Improving Generalization for Missing Data Imputation via Dual Corruption Denoising Autoencoders

## Abstract

Missing data poses challenges for machine learning applications across domains. Prevalent imputation techniques using deep learning have demonstrated limitations: GANs exhibit instability, while AutoEncoders tend to overfit. In real application scenarios, there are diverse types of missingness with varied missing rates, calling for an accurate and generic imputation approach. In this paper, we introduce Dual Corruption Denoising AutoEncoders (DC-DAE), which 1) augments inputs via dual corruptions (i.e., concurrent masking and additive noise corruptions) during training, preventing reliance on fixed missingness patterns, and enabling improved generalization; 2) applies a balanced loss function, allowing control over reconstructing artificial missingness versus denoising observed values. DC-DAE has a simple yet effective architecture without the complexity of attention mechanism or adversarial training. By combining corruption robustness and high-fidelity reconstruction, DC-DAE achieves both accuracy and stability. We demonstrate state-of-the-art performance on multiple tabular datasets with different missing rates, outperforming GAN, DAE, and VAE baselines under varied missingness scenarios. Our results highlight the importance of diverse and proper corruptions when designing models for imputation. The proposed plug-and-play approach offers an effective solution for ubiquitous missing data problems across domains.

## 1 Introduction

Missing data is a ubiquitous issue that arises across many scientific domains. Incomplete datasets are prevalent in various fields including healthcare, transportation, environmental science, and more (Wu et al. (2022), Little et al. (2012), Duan et al. (2014), Luo et al. (2019), Yoon et al. (2016), Alaa et al. (2017), Yoon et al. (2018b), Lall (2016), Wooldridge (2007)). This missing data problem can attribute to various factors including the high cost of data collection, privacy restrictions, measurement errors, system failures, and human negligence in recording information (Jaseena et al. (2014)). The presence of missing values can significantly degrade the performance of data mining algorithms and statistical models, leading to reduced predictive power, loss of statistical significance, and the introduction of bias (Camino et al. (2019)).

Numerous techniques have been advanced to mitigate the bias induced by missing data. Conventional statistical methods (e.g., mean/mode imputation, regression, Expectation-Maximization (EM), and multiple imputation) have proven effective with low missing rate scenarios (Strike et al. (2001) and Raymond & Roberts (1987)). However, these strategies tend to falter as the missing rate escalates, introducing significant bias and deviation from the authentic data distribution, thereby yielding unreliable results (Acuna & Rodriguez (2004)). This scenario has propelled the exploration of advanced solutions such as deep learning techniques, adept at discerning complex patterns and preserving data integrity, even with high missing rates.

Two major categories of deep learning imputation techniques have been explored: generative adversarial networks (GANs) and autoencoders (AEs). GAN-based approaches like Yoon et al. (2018a) treat missing data imputation as a generative modeling problem. However, GAN-based approaches often suffer from training instability and mode collapse (Salimans et al. (2016), Gulrajani et al.

(2017)). Besides, imputation differs from typical generative tasks in computer vision that require flexible latent spaces for controllable generation. In contrast, imputation is better formulated as a self-reconstruction task, since the goal is to estimate missing values consistent with the observed data distribution. Thus, AE-based methods are well-suited for this task, as they aim to encode corrupted input and reconstruct the original complete data. Variants such as denoising autoencoders (DAEs) (Vincent et al. (2008)), variational autoencoders (VAEs) (Kingma & Welling (2013)), and overcomplete AEs (Gondara & Wang (2018)) have been applied for missing data imputation. Nevertheless, VAEs make strong assumptions about latent variable distributions that may not match real data. Standard DAEs fail to generalize to samples with missingness patterns not seen during training. Overcomplete AEs require large networks that are resource intensive and could allow overfitting.

To overcome the limitations of prior techniques, we introduce Dual Corruption Denoising Autoencoders (DC-DAE) - a novel autoencoder-based approach that concurrently injects masking and additive noise corruptions during training. DC-DAE provides several key advantages:

- The dual corruptions prevent overreliance on any fixed missingness pattern by exposing the model to diverse missingness. This facilitates generalization to new missing data distributions.

- The balanced loss function allows configurable tradeoff between reconstructing artificial missingness and denoising observed values. This is tailored for robust performance across heterogeneous real-world datasets.

- DC-DAE utilizes a lightweight architecture without overparameterization. And our empirical results demonstrate state-of-the-art imputation accuracy across a wide range of missingness scenarios, outperforming prior AE variants.

The rest of the paper is organized as follows: Section 2 reviews related work across GAN and AE based deep learning techniques for missing data imputation. Section 3 provides the problem formulation. Section 4 explains our proposed DC-DAE methodology. Section 5 describes our experimental setup, results, and analysis. Section 6 discusses limitations and future work. Finally, Section 7 concludes with a summary of our contributions.

## 2 Related Work

A variety of deep learning techniques have been proposed for missing data imputation and can be categorized into two main approaches:

### 2.1 Generative Adversarial Network (GAN) based Techniques

Generative adversarial networks (GANs) have been explored for missing data imputation by formulating it as a generative modeling problem (e.g., GAIN, Yoon et al. (2018a)). The generator creates synthetic data and the discriminator tries to distinguish real versus generated data (Goodfellow et al. (2014)). However, GANs often suffer from issues like mode collapse and training instability (Salimans et al. (2016), Gulrajani et al. (2017)).

Approaches aim to improve GAN-based imputation through attention layers (e.g., SA-GAIN, Zhang et al. (2021)), additional discriminators (e.g., MisGAN, Li et al. (2019)), temporal architectures (e.g., E2GAN, Luo et al. (2019)), auxiliary tasks (e.g., HexaGAN, Hwang et al. (2019)), and skip connections (e.g., ST-LBAGAN, Yang et al. (2021)). These enhance sample quality and training stability to some extent. However, fundamental challenges remain in properly balancing generator and discriminator capacities over training (Hwang et al. (2019)).

Moreover, the flexibility of generative modeling may exceed the requirements of reconstructive imputation (Yoon & Sull (2020)). Latent spaces tailored for controllable generation are likely unnecessary or even detrimental. The instability and excessive modeling flexibility make GANs less than ideal for missing data imputation.

## 2.2 Autoencoder (AE) based Techniques

AutoEncoders (AEs) are a natural fit for missing data imputation, as they aim to reconstruct corrupted inputs. The encoder maps the input data to a latent representation, and the decoder attempts to reconstructs the original input from this compressed code.

For imputation, the model is trained to fill in missing values in order to minimize the reconstruction error between the input and output. However, traditional AEs often struggle to generalize to missingness patterns not seen during training (Gondara & Wang (2018), Tihon et al. (2021)). They may overfit to the specific fixed missingness in the training data distribution, failing to effectively estimate values for new missingness patterns.

A variety of techniques have been proposed to enhance the imputation accuracy and generalization of AEs, such as OAE/DAE/VAE-based methods :

- Overcomplete AEs (i.e., OAE, Gondara & Wang (2018)) use more hidden units than inputs, improving representation capacity.

- Denoising AEs (Vincent et al. (2008)) inject noise or missingness during training as an explicit corruption process. The model learns to reconstruct the original undistorted input.

- Stacked DAEs (Sánchez-Morales et al. (2020), Costa et al. (2018)) ensemble multiple DAEs for improved modeling. Dropout regularization further enhances generalization.

- Variational AEs (i.e., VAE, Kingma & Welling (2013), e.g., HI-VAE, Nazabal et al. (2020) and MIWAE, Mattei & Frellsen (2019)) impose assumptions on the latent space distribution and enable generative sampling.

- Attention-based AEs ( e.g., AimNet, Wu et al. (2020) and DAEMA, Tihon et al. (2021)) allow the model to focus on relevant input features when reconstructing mixed data.

However, fundamental limitations still remain. E.g., Latent space assumptions in VAEs may not match real data. Standard AEs fail to generalize(Gondara & Wang (2018)) , while overcomplete networks require extensive resources. Moreover, added complexity from attention does not guarantee benefits. Thus, more advanced regularization and architecture design are needed to improve robustness and generalization of AEs for missing data imputation.

Some recent approaches like IM-GAN (Wu et al. (2022)) utilizes a bidirectional RNN, DAE, and GAN for time series data imputation, learning temporal correlations. Note, this demonstrates potential benefits of complementing AEs with GAN components, however, stability and performance gains are not guaranteed (Yoon & Sull (2020)).

To address these challenges, we propose a novel yet simple approach called Dual Corruption Denoising Autoencoder (DC-DAE). DC-DAE augments the input data during training via concurrent corruption with masking and noise. Additionally, we apply different loss weights to the masking reconstruction task versus the denoising reconstruction task. We demonstrate that this rebalancing prevents overfitting to any one fixed missingness mechanism, facilitating generalization. The lightweight architecture requires no adversarial training, overcomplete representations, or attention modules. Our experiments across diverse datasets and missingness scenarios demonstrate state-of-the-art performance compared to existing imputation techniques. The straightforward reconstructive framework of DC-DAE provides an effective and efficient solution for missing data imputation.

## 3 Problem Formulation

Consider a dataset $\mathbb{D} = \{\boldsymbol{X}\}$. $\boldsymbol{X} \in \mathbb{R}^{n \times d}$ is a matrix composed of $n$ samples and each sample is composed by $d$ features: $\boldsymbol{x}_i = (x_i^1, x_i^2, \ldots, x_i^d)$. As for the missing data imputation problem, a dataset with natural missing values in is given as $\boldsymbol{X}$. We denote $\boldsymbol{X}^*$ the ground truth dataset without missingness. We define a new data matrix $\widetilde{\boldsymbol{X}} \in \mathbb{R}^{n \times d}$ and mask matrix $\boldsymbol{M} \in \mathbb{R}^{n \times d}$ in the following way:

$$\widetilde{x}_i^j = \left\{ \begin{array}{ll} x_i^j & (m_i^j = 0) \\ 0 & (m_i^j = 1) \end{array} \right. \tag{1}$$

$$m_i^j = \begin{cases} 0 & (observed\ data) \\ 1 & (natural\ missing\ data) \end{cases} \tag{2}$$

Our goal is to impute all the missing components of $\boldsymbol{X}$ and make it as close to $\boldsymbol{X}^*$ as possible. Formally, we want to obtain imputed data matrix $\hat{\boldsymbol{X}} \in \mathbb{R}^{n \times d}$ given $\widetilde{\boldsymbol{X}}$ and $\boldsymbol{M}$:

$$\hat{\boldsymbol{X}} = f(\widetilde{\boldsymbol{X}}, \boldsymbol{M}) \tag{3}$$

where f is the imputation method.

## 4 METHODOLOGY

### 4.1 OVERALL ARCHITECTURE

As shown in Figure 1, the DC-DAE framework consists of two key phases: 1) concurrent corruption of the input data with masking and noise, and 2) reconstruction of the original complete data using a DAE architecture.

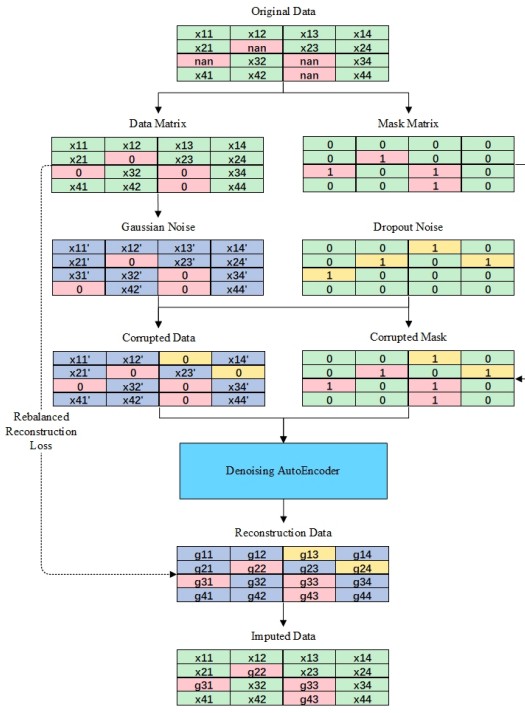

Figure 1: DC-DAE architecture overview

### 4.2 MASKED DENOISING

A key innovation of DC-DAE is jointly utilizing two different corruption techniques during training:

- Noise Injection: Adding Gaussian random noise $\epsilon$ to the observed values. The noise scale hyperparameter $\gamma$ determines the noise level. $\epsilon \sim \mathcal{N}(0, \gamma)$

$$\dot{\boldsymbol{X}} = \widetilde{\boldsymbol{X}} + \epsilon * (1 - \boldsymbol{M}) \tag{4}$$

- Masking: Randomly dropout, or "masking", a subset of the observed feature values to create artificial missingness. The mask ratio hyperparameter $\beta$ controls the amount of masking. Let $\overline{\boldsymbol{M}}$ be the artificial mask matrix, then

$$\ddot{\boldsymbol{X}} = \dot{\boldsymbol{X}} * (1 - \overline{\boldsymbol{M}}) \tag{5}$$

Here, * operation denotes element-wise multiplication. Applying these corruptions prevents the model from directly accessing clean observed data. Additionally, it exposes the model to diverse missingness during training, improving generalization to unseen patterns at test time.

## 4.3 RECONSTRUCTION

The corrupted input then goes through a DAE to reconstruct the original complete data. DAE is trained to invert the effect of corruption by mapping corrupted samples to clean samples.

$$\overline{X} = \hat{f}(\ddot{X}, \overline{M} * M) \tag{6}$$

$\hat{f}$ is the reconstruction network, which architecture consists of 3 encoder layers which map the input to a latent representation, followed by 3 symmetric decoder layers that reconstruct the output from the code. All layers use tanh activation and the dimensionality of the hidden layers matches the input feature dimension. Combining with observed data, DC-DAE final imputation results are:

$$\hat{X} = \overline{X} * M + \widetilde{X} * (1 - M) \tag{7}$$

## 4.4 LOSS FUNCTION

The overall reconstruction loss is defined as:

$$L = \alpha * L_n + (1 - \alpha) * L_m \tag{8}$$

$$L_n = ||(\overline{X} - \widetilde{X}) * (1 - \overline{M}) * (1 - M)|| \tag{9}$$

$$L_m = ||(\overline{X} - \widetilde{X}) * \overline{M} * (1 - M)|| \tag{10}$$

Here, $L_m$ penalizes reconstruction error on the artificially masked entries, while $L_n$ penalizes error on noising observed values. The $\alpha$ parameter controls the balance between the two objectives. Setting $\alpha < 0.5$ puts less emphasis on reconstructing the noisy observed data. We demonstrate experimentally that this rebalancing improves imputation accuracy by preventing overfitting to the artificial masking distribution. The loss hyperparameters are tuned on a validation set.

# 5 EXPERIMENTS

## 5.1 DATASETS

We evaluate DC-DAE on 5 real-world tabular datasets from the UCI repository (Kelly et al.). As is shown in Table 1, these datasets cover a diverse range of sizes and dimensions to comprehensively evaluate performance between DC-DAE and various state-of-the-art imputation methods. We remove the ID column in the Glass and Breast datasets and samples contain NA values in Breast dataset.

Table 1: Datasets used for evaluation

| Dataset Full Name | Dataset Acronym | Samples | Features |
|---|---|---|---|
| Glass Identification | Glass | 214 | 9 |
| Breast Cancer Wisconsin (Original) | Breast | 683 | 9 |
| Spambase | Spam | 4601 | 57 |
| Letter Recognition | Letter | 20000 | 16 |
| Statlog (Shuttle) | Shuttle | 58000 | 8 |

## 5.2 EVALUATION PROTOCOL

We introduce missingness artificially via Missing Completely at Random (MCAR) masking[1]. The datasets are split 70/30 into train/test. Missing values in the test set are imputed and evaluated.

---

[1]We also evaluate performance under Missing Not at Random (MNAR) missingness in the Appendix A.1

Normalization uses train set statistics. Performance is measured by normalized root mean squared error (NRMSE) on the missing test values across 10 random missingness trials. We report average NRMSE over these trials. NRMSE is a common imputation evaluation metric used by baseline methods like DAEMA.

## 5.3 IMPLEMENTATION DETAILS

The hyperparameters are set as:

- Mask ratio $\beta$: set to be 0.2 based on common usage in baseline methods.
- Noise scale $\gamma$: set to be 0.2 for simplicity.
- Denoising loss weight $\alpha$: Tuned on validation data across the range 0.1 to 0.9. Optimal values found were 0.1 (Breast, Spam), 0.2 (Letter, Shuttle), 0.9 (Glass).

The model architecture uses 6 fully-connected layers with tanh activations and hidden dimensionality matching the input features. We select 6 layers to match the default depth required by the MIDA baseline method and to enable fair comparison by allowing other baselines to use the same capacity if desired. Models are trained with batch size 64 for 10000 epochs, using the Adam optimizer with a learning rate of 0.001.

## 5.4 BASELINE

We compare DC-DAE with several state-of-the-art imputation methods:

- Traditional: Mean, MICE (Van Buuren & Groothuis-Oudshoorn (2011))
- GAN-based: GAIN (Yoon et al. (2018a)), IM-GAN (Wu et al. (2022))
- DAE-based: MIDA (Gondara & Wang (2018)), DAEMA (Tihon et al. (2021))
- VAE-based: HI-VAE (Nazabal et al. (2020)), MIWAE (Mattei & Frellsen (2019))

Recent works (Camino et al. (2020), Lin & Tsai (2020)) have identified challenges in reproducing and fairly comparing deep learning imputation algorithms due to inconsistencies in evaluation protocols and difference in model capacity. To mitigate this, we implement a standardized experiment procedure across all algorithms. The same train/test splits, missingness patterns, and evaluation metrics are applied to each method. Additionally, baseline architectures are adjusted to match the capacity of DC-DAE where possible, while retaining their core characteristics:

- For GAIN, we use equal hidden layer sizes rather than bottleneck layers and construct a 6-layer generator and a 3-layer discriminator.
- For IM-GAN, we replace BRNN cells with fully-connected networks in the same 6+3 architecture as GAIN.
- For DAEMA, we reduce the hidden size from [2d, 2d] to [d, 1] to prevent overparameterization and set the depth of feature decoder to 3.
- For MIWAE and HI-VAE, the decoder depth is set to 3 layers.
- For MIDA, we use the default hyperparameters from the original papers.
- For MICE, we utilize the IterativeImputer module from the scikit-learn Python package with its default settings.

This standardized protocol enables direct comparison between DC-DAE and state-of-the-art baselines.

## 5.5 RESULTS

The experimental results show that the proposed DC-DAE method has achieved superior imputation performance compared to existing deep learning techniques. This validates the author's hypothesis that the dual corruption technique and balanced loss can enhance the generalization ability for various missing patterns. The results are consistent with the design objectives, with DC-DAE achieving

Table 2: Imputation performance (NRMSE) comparison between DC-DAE and baseline methods

| | Glass | Breast | Spam | Letter | Shuttle |
|---|---|---|---|---|---|
| | Low miss rate: $beta = 0.2$ | | | | |
| Mean | $1.003_{\pm0.112}$ | $0.993_{\pm0.035}$ | $1.007_{\pm0.039}$ | $0.997_{\pm0.005}$ | $0.969_{\pm0.077}$ |
| MICE | $0.757_{\pm0.152}$ | $0.679_{\pm0.037}$ | $1.051_{\pm0.188}$ | $0.746_{\pm0.006}$ | $\mathbf{0.573}_{\pm0.121}$ |
| GAIN | $1.221_{\pm0.277}$ | $1.298_{\pm0.078}$ | $1.196_{\pm0.034}$ | $1.297_{\pm0.041}$ | $0.928_{\pm0.093}$ |
| IM-GAN | $1.464_{\pm0.330}$ | $1.291_{\pm0.694}$ | $0.980_{\pm0.040}$ | $0.994_{\pm0.031}$ | $0.830_{\pm0.088}$ |
| MIDA | $0.798_{\pm0.090}$ | $0.736_{\pm0.031}$ | $0.942_{\pm0.038}$ | $0.802_{\pm0.006}$ | $0.802_{\pm0.091}$ |
| DAEMA | $\mathbf{0.739}_{\pm0.093}$ | $0.685_{\pm0.050}$ | $0.950_{\pm0.040}$ | $0.709_{\pm0.010}$ | $0.595_{\pm0.117}$ |
| HI-VAE | $0.963_{\pm0.179}$ | $0.766_{\pm0.042}$ | $1.027_{\pm0.040}$ | $0.750_{\pm0.011}$ | $0.773_{\pm0.100}$ |
| MIWAE | $0.903_{\pm0.125}$ | $0.759_{\pm0.047}$ | $1.014_{\pm0.044}$ | $0.742_{\pm0.028}$ | $0.674_{\pm0.122}$ |
| DC-DAE(ours) | $0.788_{\pm0.118}$ | $\mathbf{0.661}_{\pm0.041}$ | $\mathbf{0.927}_{\pm0.040}$ | $\mathbf{0.669}_{\pm0.006}$ | $0.595_{\pm0.118}$ |
| | Medium miss rate: $beta = 0.5$ | | | | |
| Mean | $0.998_{\pm0.089}$ | $1.007_{\pm0.020}$ | $0.993_{\pm0.029}$ | $0.999_{\pm0.003}$ | $0.982_{\pm0.040}$ |
| MICE | $1.130_{\pm0.168}$ | $0.843_{\pm0.070}$ | $1.225_{\pm0.200}$ | $0.856_{\pm0.009}$ | $0.934_{\pm0.135}$ |
| GAIN | $1.534_{\pm0.470}$ | $1.285_{\pm0.124}$ | $1.113_{\pm0.030}$ | $1.378_{\pm0.050}$ | $1.256_{\pm0.065}$ |
| IM-GAN | $2.267_{\pm0.605}$ | $1.469_{\pm0.309}$ | $1.036_{\pm0.027}$ | $1.062_{\pm0.025}$ | $1.041_{\pm0.122}$ |
| MIDA | $0.958_{\pm0.060}$ | $0.863_{\pm0.026}$ | $\mathbf{0.966}_{\pm0.026}$ | $0.934_{\pm0.003}$ | $0.924_{\pm0.043}$ |
| DAEMA | $0.965_{\pm0.079}$ | $0.731_{\pm0.033}$ | $0.993_{\pm0.026}$ | $0.804_{\pm0.005}$ | $0.737_{\pm0.052}$ |
| HI-VAE | $0.943_{\pm0.083}$ | $0.824_{\pm0.049}$ | $1.015_{\pm0.029}$ | $0.853_{\pm0.008}$ | $0.857_{\pm0.043}$ |
| MIWAE | $\mathbf{0.940}_{\pm0.093}$ | $0.812_{\pm0.036}$ | $1.006_{\pm0.031}$ | $0.832_{\pm0.026}$ | $0.883_{\pm0.047}$ |
| DC-DAE(ours) | $0.942_{\pm0.084}$ | $\mathbf{0.717}_{\pm0.053}$ | $0.970_{\pm0.036}$ | $\mathbf{0.786}_{\pm0.006}$ | $\mathbf{0.730}_{\pm0.055}$ |
| | High miss rate: $beta = 0.8$ | | | | |
| Mean | $1.035_{\pm0.058}$ | $1.006_{\pm0.014}$ | $1.000_{\pm0.014}$ | $1.000_{\pm0.002}$ | $0.991_{\pm0.018}$ |
| MICE | $1.280_{\pm0.202}$ | $1.010_{\pm0.052}$ | $1.185_{\pm0.221}$ | $0.983_{\pm0.012}$ | $1.280_{\pm0.430}$ |
| GAIN | $1.736_{\pm0.578}$ | $1.386_{\pm0.057}$ | $1.105_{\pm0.015}$ | $1.365_{\pm0.033}$ | $1.303_{\pm0.022}$ |
| IM-GAN | $3.233_{\pm0.709}$ | $2.349_{\pm0.377}$ | $1.499_{\pm0.124}$ | $2.264_{\pm0.285}$ | $2.325_{\pm0.613}$ |
| MIDA | $\mathbf{1.034}_{\pm0.052}$ | $0.977_{\pm0.015}$ | $\mathbf{0.995}_{\pm0.013}$ | $0.990_{\pm0.002}$ | $0.983_{\pm0.017}$ |
| DAEMA | $1.272_{\pm0.189}$ | $0.861_{\pm0.023}$ | $1.086_{\pm0.035}$ | $0.920_{\pm0.003}$ | $0.904_{\pm0.019}$ |
| HI-VAE | $\mathbf{1.034}_{\pm0.034}$ | $0.910_{\pm0.035}$ | $1.024_{\pm0.015}$ | $0.965_{\pm0.025}$ | $0.967_{\pm0.029}$ |
| MIWAE | $1.042_{\pm0.034}$ | $0.887_{\pm0.030}$ | $1.023_{\pm0.016}$ | $0.931_{\pm0.002}$ | $0.966_{\pm0.024}$ |
| DC-DAE(ours) | $1.191_{\pm0.162}$ | $\mathbf{0.839}_{\pm0.029}$ | $1.029_{\pm0.015}$ | $\mathbf{0.914}_{\pm0.003}$ | $\mathbf{0.893}_{\pm0.019}$ |

a balance of accuracy and efficiency. It demonstrates the state-of-the-art performance of our proposed DC-DAE method against existing deep learning techniques for missing data imputation. Furthermore, the baseline comparisons in Table 2 highlight that DC-DAE achieves lower NRMSE than GAN-based, AE-based, and VAE-based approaches across diverse datasets and missingness ratios.

## 5.6 ANALYSIS

The experimental results validate the efficacy of the proposed DC-DAE technique and provide insights into the sources of performance gains. We conduct in-depth ablation studies on the Letter dataset, systematically evaluating the contribution of each component of our approach (Figure 2). The different sample sizes show ablation experiments using subsets of the full dataset, which evaluates model performance according to the availability of training data. The figures report performance as a percentage of NRMSE compared to the corresponding baseline, with lower percentages indicating improvement.

### 5.6.1 SOURCE OF GAIN

Introducing artificial missingness via random observation masking consistently improves imputation accuracy over a vanilla AE across datasets, sample sizes, and missingness ratios (Figure 2a). This indicates that training on both real and synthetic missing values, which the model cannot differentiate, regularizes the model and prevents overfitting to any one fixed pattern. Further corrupting the remaining observed data with Gaussian noise provides additional gains in the majority of settings (Figure 2b). By preventing direct access to clean observations, the model must reconstruct the orig-

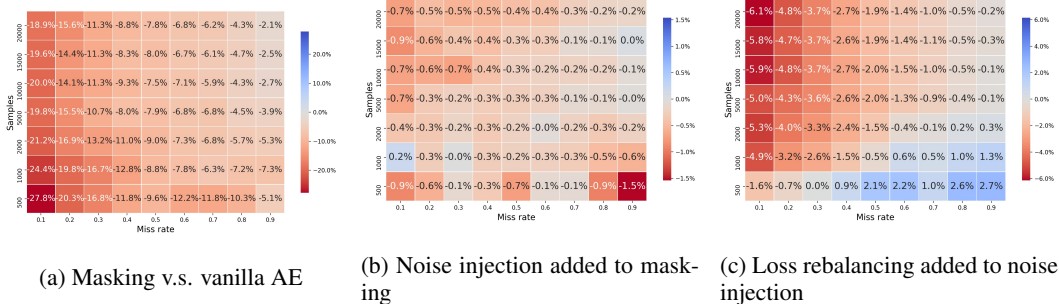

Figure 2: Incremental ablation studies showing gains from (a) masking, (b) adding noise, and (c) rebalancing loss, relative to previous step. The tables report performance as a percentage of NRMSE compared to the corresponding baseline, with lower percentages indicating improvement.

inal values from the noisy inputs during training. This denoising task acts as an effective auxiliary regularizer. Tuning the loss rebalancing hyperparameter $\alpha$ allows control over the relative importance of reconstructing the artificially introduced missingness versus denoising the randomly noised observed data. As shown in Figure 2c, this prevents overfitting to the noised observation distribution, especially for large samples and low missing rates where noised observed data dominates.

### 5.6.2 ATTENTION AND GAN

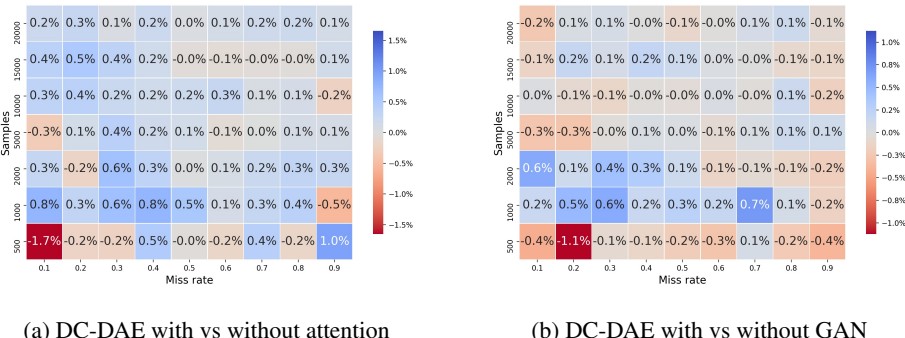

Figure 3: DC-DAE with (a) attention and (b) GAN components

Incorporating cross-attention between the data and mask provides minimal benefits, even decreasing performance in some scenarios (Figure 3a). This indicates that the additional complexity is unneeded for DC-DAE's straightforward reconstructive approach. Similarly, introducing an adversarial discriminator offers gains only in some scenarios (Figure 3b). This aligns with the finding that flexible generative modeling is unnecessary for the primary task of missing value imputation.

### 5.6.3 OVERCOMPLETE

Expanding the hidden layer sizes approximating MIDA and VAE architectures consistently improves results, especially for low missingness (Figure 4). This highlights the value of increased representational capacity despite DC-DAE's lightweight architecture. Integrating overcomplete representations could further improve performance.

In summary, the analysis verifies that the dual corruption techniques and balanced loss formulation are the primary drivers of DC-DAE's effectiveness. The results provide insights into appropriate inductive biases for imputation.

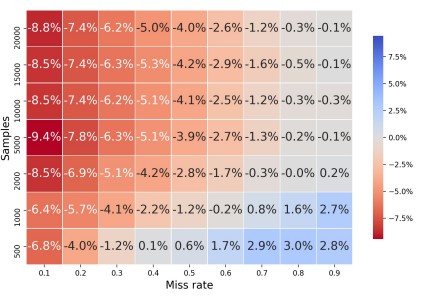 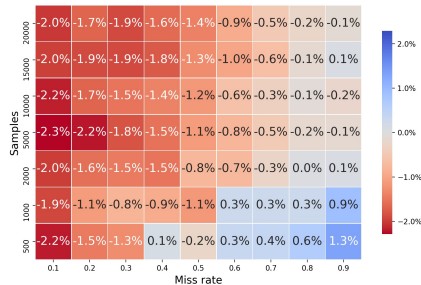

(a) DC-DAE with MIDA overcomplete representation

(b) DC-DAE with VAE overcomplete representation

Figure 4: DC-DAE with overcomplete representations approximating (a) MIDA and (b) VAEs

# 6 FUTURE WORK

While the proposed DC-DAE method achieves state-of-the-art imputation accuracy, several promising directions remain for future work:

- As the auxiliary denoising task provides benefits, exploring more advanced multi-task learning approaches like MMOE (Ma et al. (2018)) could further improve performance.
- Applying more sophisticated methods to handle mixed data types with both numerical and categorical features could enhance applicability to broader tabular dataset
- Tuning the hyperparameters like mask ratio, nose scale and denoising loss weight simultaneously in a dataset-specific or automated manner could improve robustness across diverse missing data scenarios.
- Incorporating overcomplete representations and greater model capacity could potentially build upon the benefits of the masking denoising technique.

By addressing these limitations and extensions, the power of the simple yet effective masked denoising approach could be further leveraged to tackle missing data challenges.

# 7 CONCLUSIONS

This work introduced Dual Corruption Denoising Autoencoders (DC-DAE), a straightforward yet powerful technique for missing data imputation. DC-DAE corrupts input data during training through simultaneous masking and additive noise. We demonstrate experimentally that this prevents overfitting to any one artificial missingness pattern, facilitating imputation under real-world missingness conditions at test time.

The balanced loss function further enhances the approach by controlling the tradeoff between reconstructing artificial missingness and denoising observed values. Our experiments across diverse datasets and missingness scenarios demonstrate state-of-the-art performance compared to existing deep learning imputation methods based on GANs, AEs, and VAEs.

The plug-and-play masked denoising technique provides an intuitive and effective solution for handling missing data. By combining simple corruption robustness with reconstructive modeling, DC-DAE achieves both accuracy and efficiency. The strong empirical results highlight the value of proper inductive bias when designing deep learning architectures for missing data problems.

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

# A    APPENDIX

## A.1    ADDITIONAL EXPERIMENTS ON MISSING NOT AT RANDOM (MNAR) DATA

Missing data which arises under different missingness mechanisms can be categorized as Missing Completely at Random (MCAR), Missing at Random (MAR), and Missing Not at Random (MNAR) (Baraldi & Enders (2010), Rubin (1976), Little & Rubin (2019)). Under MCAR, the probability of a data point being missing is unrelated to the values of any features. For MAR, the missingness probability depends only on the observed data rather than the missing values. And under MNAR, the missingness depends on the missing values themselves. The distribution of real-world missing data can fall under any of these mechanisms.

In the main paper, we focused our experiments and analysis on MCAR missing data. Here, we provide additional results validating the performance of DC-DAE on data with MNAR missingness.

To evaluate performance under MNAR missingness, we generate the missing data as follows:

1. Randomly select 2 features from the dataset
2. Select samples where:
   - Feature 1 $\leq$ median of feature 1
   - Feature 2 $\geq$ median of feature 2
3. For only those selected samples, randomly remove 20% of values across all features.

This results in missingness that depends on the value distribution of the 2 chosen features, but is broadly applied across features for those samples meeting the criteria. The same evaluation protocol as the main MCAR experiments is then applied.

As shown in Table 3, DC-DAE continues to achieve strong performance under MNAR across the datasets and across low (0.2), medium (0.5) and high (0.8) missingness ratios. Specifically, DC-DAE outperforms all baseline methods on 3 out of 5 datasets when missing rate is 0.2, 4 out of 5 datasets when missing rate is 0.5, and all 5 datasets when missing rate is 0.8.

Table 3: Imputation performance (NRMSE) comparison between DC-DAE and baseline methods

| | Glass | Breast | Spam | Letter | Shuttle |
|---|---|---|---|---|---|
| | Low miss rate: $beta = 0.2$ | | | | |
| Mean | $0.957_{\pm0.119}$ | $1.018_{\pm0.041}$ | $1.012_{\pm0.051}$ | $0.990_{\pm0.011}$ | $1.027_{\pm0.148}$ |
| MICE | $0.762_{\pm0.205}$ | $0.667_{\pm0.049}$ | $1.120_{\pm0.393}$ | $0.737_{\pm0.012}$ | $\mathbf{0.649}_{\pm0.230}$ |
| GAIN | $1.208_{\pm0.281}$ | $1.215_{\pm0.090}$ | $1.191_{\pm0.048}$ | $1.268_{\pm0.050}$ | $0.950_{\pm0.203}$ |
| IM-GAN | $1.246_{\pm0.434}$ | $1.111_{\pm0.088}$ | $0.984_{\pm0.046}$ | $0.969_{\pm0.023}$ | $0.801_{\pm0.206}$ |
| MIDA | $0.783_{\pm0.121}$ | $0.757_{\pm0.061}$ | $0.939_{\pm0.051}$ | $0.790_{\pm0.008}$ | $0.856_{\pm0.177}$ |
| DAEMA | $0.737_{\pm0.124}$ | $0.682_{\pm0.047}$ | $0.948_{\pm0.049}$ | $0.706_{\pm0.012}$ | $0.677_{\pm0.224}$ |
| HI-VAE | $0.863_{\pm0.125}$ | $0.781_{\pm0.063}$ | $1.033_{\pm0.050}$ | $0.749_{\pm0.013}$ | $0.843_{\pm0.190}$ |
| MIWAE | $0.899_{\pm0.154}$ | $0.765_{\pm0.059}$ | $1.020_{\pm0.051}$ | $0.723_{\pm0.012}$ | $0.726_{\pm0.228}$ |
| DC-DAE | $\mathbf{0.721}_{\pm0.122}$ | $\mathbf{0.662}_{\pm0.043}$ | $\mathbf{0.926}_{\pm0.047}$ | $\mathbf{0.674}_{\pm0.009}$ | $0.677_{\pm0.223}$ |
| | Medium miss rate: $beta = 0.5$ | | | | |
| Mean | $0.975_{\pm0.090}$ | $1.008_{\pm0.034}$ | $0.999_{\pm0.015}$ | $0.998_{\pm0.009}$ | $0.984_{\pm0.061}$ |
| MICE | $0.970_{\pm0.191}$ | $0.822_{\pm0.035}$ | $1.202_{\pm0.114}$ | $0.866_{\pm0.025}$ | $0.827_{\pm0.070}$ |
| GAIN | $1.309_{\pm0.211}$ | $1.397_{\pm0.120}$ | $1.124_{\pm0.025}$ | $1.362_{\pm0.036}$ | $1.214_{\pm0.130}$ |
| IM-GAN | $2.168_{\pm0.401}$ | $1.437_{\pm0.193}$ | $1.067_{\pm0.053}$ | $1.076_{\pm0.033}$ | $0.946_{\pm0.127}$ |
| MIDA | $0.904_{\pm0.076}$ | $0.865_{\pm0.038}$ | $\mathbf{0.965}_{\pm0.019}$ | $0.918_{\pm0.006}$ | $0.909_{\pm0.066}$ |
| DAEMA | $0.882_{\pm0.120}$ | $0.729_{\pm0.030}$ | $1.000_{\pm0.030}$ | $0.805_{\pm0.010}$ | $0.734_{\pm0.082}$ |
| HI-VAE | $0.923_{\pm0.082}$ | $0.805_{\pm0.040}$ | $1.020_{\pm0.014}$ | $0.841_{\pm0.016}$ | $0.856_{\pm0.079}$ |
| MIWAE | $0.902_{\pm0.099}$ | $0.793_{\pm0.036}$ | $1.013_{\pm0.020}$ | $0.810_{\pm0.023}$ | $0.852_{\pm0.069}$ |
| DC-DAE | $\mathbf{0.853}_{\pm0.115}$ | $\mathbf{0.712}_{\pm0.032}$ | $0.970_{\pm0.029}$ | $\mathbf{0.785}_{\pm0.009}$ | $\mathbf{0.729}_{\pm0.081}$ |
| | High miss rate: $beta = 0.8$ | | | | |
| Mean | $1.036_{\pm0.097}$ | $1.018_{\pm0.023}$ | $0.998_{\pm0.012}$ | $1.019_{\pm0.013}$ | $0.991_{\pm0.056}$ |
| MICE | $1.095_{\pm0.210}$ | $0.925_{\pm0.049}$ | $1.232_{\pm0.262}$ | $0.969_{\pm0.016}$ | $1.029_{\pm0.100}$ |
| GAIN | $1.394_{\pm0.190}$ | $1.382_{\pm0.105}$ | $1.118_{\pm0.020}$ | $1.370_{\pm0.047}$ | $1.290_{\pm0.049}$ |
| IM-GAN | $2.968_{\pm0.677}$ | $2.569_{\pm0.422}$ | $1.484_{\pm0.135}$ | $2.112_{\pm0.323}$ | $2.190_{\pm0.748}$ |
| MIDA | $\mathbf{1.012}_{\pm0.103}$ | $0.984_{\pm0.020}$ | $\mathbf{0.992}_{\pm0.013}$ | $0.997_{\pm0.011}$ | $0.967_{\pm0.054}$ |
| DAEMA | $1.047_{\pm0.159}$ | $0.879_{\pm0.021}$ | $1.096_{\pm0.037}$ | $0.919_{\pm0.008}$ | $0.873_{\pm0.061}$ |
| HI-VAE | $1.014_{\pm0.096}$ | $0.911_{\pm0.034}$ | $1.022_{\pm0.011}$ | $0.944_{\pm0.014}$ | $0.943_{\pm0.063}$ |
| MIWAE | $1.019_{\pm0.096}$ | $0.888_{\pm0.031}$ | $1.020_{\pm0.014}$ | $0.922_{\pm0.014}$ | $0.941_{\pm0.059}$ |
| DC-DAE | $1.036_{\pm0.128}$ | $\mathbf{0.857}_{\pm0.036}$ | $1.045_{\pm0.031}$ | $\mathbf{0.910}_{\pm0.008}$ | $\mathbf{0.867}_{\pm0.061}$ |

