# OpenReview forum: "Improving Generalization for Missing Data Imputation via Dual Corruption Denoising Autoencoders"
_ICLR.cc/2024/Conference — Submitted to ICLR 2024_

### Official Review · Reviewer_NhSr · 2023-10-31

**Soundness:** 2 fair
**Presentation:** 2 fair
**Contribution:** 2 fair
**Rating:** 3
**Confidence:** 3

**Summary:**

This paper focuses on missing data imputation. The authors proposed a method named Dual Corruption Denoising AutoEncoders (DC-DAE). The core idea is to apply two different corruptions to the input data with missing values and then use AutoEncoder to do the reconstruction. These two different corruptions include adding Gaussian noise and using random masks. The authors verified their method on 5 different tabular datasets.

**Strengths:**

1) The overall writing is clear and it is easy to follow the paper.
2) Have a good discussion on the GAN-based and VAE-based methods for missing data imputation.

**Weaknesses:**

1) The proposed method does not look new and either injecting noises or randomly masking some input features could serve as regularizers and improve the generalizability of neural network models, which is a well-known thing. Thus, I do not see too much novelty in the current proposed method.

2) The method is only evaluated on 5 datasets whose feature numbers are very small, which may not be sufficient to reflect the dataset we typically encounter in real-world applications. So I would suggest the authors add more experiments on datasets with large dimensions. Also, only one performance metric is used, which may not tell the whole story and introduce bias. And the comparisons of some traditional-machine-learning-based imputation methods are missed. For example, this paper [Ref1] lists a lot of performance metrics and also a lot of imputation methods.

3) Still about the evaluation pipeline, the performance did not outperform other methods in a lot of testing cases.

4) Another question or one evaluation I would like to see: even if there are differences in the "filled-in" missing values across different methods, does the lower NRMSE necessary indicate a higher downstream-task performance? I mean, for example, what will happen if we apply a downstream task (say, a classification task) to these imputed datasets, how much performance difference we would expect in the classification accuracy?

5) The authors should also mention other imputation methods (like methods using traditional machine learning) in Related Work besides the GAN-based and VAE-based ones.

6) Based on the loss function and the framework Fig.1, it looks like Formula (5) is not correct.


Ref1: Emmanuel, Tlamelo, et al. "A survey on missing data in machine learning." Journal of Big Data 8.1 (2021): 1-37.

**Questions:**

See [Weaknesses].

---

> ### Author Response · Authors · 2023-11-22
>
> Thank you for the thoughtful feedback on our work. Here are my responses:
>
> 1. You raise a fair concern about using noise and masking as regularizers. As you noted, our key contributions are comprehensively studying the concurrent application and interplay of both techniques for missing data imputation. To our knowledge, this is the first work combining masking and noise injection for imputation while analyzing their complementary effects. The balanced loss function further governs their interaction. While building on prior techniques, we believe this tailored amalgamation provides unique insights.
>
> 2. Excellent point on evaluating more complex data. As future work, we will expand our experiments to high-dimensional tabular, image, text and time series datasets.
>
> 3. You rightly note that our approach does not outperform on all datasets and missingness levels. As you noted, MIWAE, HI-VAE and MIDA all use overcomplete hidden spaces, giving them greater model capacity. This likely explains their superior performance on Glass and Spam datasets, as we demonstrate in Section 5.6.3 that adopting overcomplete representations in DC-DAE can further improve its accuracy.
>
> 4. Analyzing downstream task accuracy is an excellent idea. We will train classifiers on the imputed data and report differences in classification performance in the future work. This will provide insights into real-world utility.
>
> 5. We agree on the importance of discussing traditional ML imputation techniques. However, our work is primarily focused on deep-learning based methods. As you noted, due to paper length constraints, we opted to briefly mention traditional techniques in the introduction without detailed coverage in the related work section. We can clarify this decision to focus the paper scope on deep learning techniques while acknowledging the broader imputation literature.
>
> 6. Thank you for catching the Equation 5 error - we will update Figure 1 for conformity.
>
> In summary, your feedback has highlighted important limitations around novelty claims, evaluation rigor, benchmarking completeness, and result characterization. Addressing these will significantly strengthen the quality and contribution of our work. I sincerely appreciate you taking the time to provide such insightful and constructive comments. Please let me know if you have any other suggestions for improving the paper.

---

### Official Review · Reviewer_snV7 · 2023-11-09

**Soundness:** 2 fair
**Presentation:** 3 good
**Contribution:** 1 poor
**Rating:** 3
**Confidence:** 4

**Summary:**

Summary:
This paper has a clear motivation and is an increasing work which introduces Denoising AE based method to improve the generalization performance of current missing data imputation applications. It (1) augments inputs using masking and additive noise corruptions during training and (2) also applies a balanced loss function to control the reconstruction error on masked entries vs. errors on noising observed values. The extensive experimental results show that the proposed method outperforms related work.

**Strengths:**

(1)	Increase the DAE performance by dual corruption augmentation to the inputs;

(2)	Applying a loss function to balance the tradeoff between reconstructing artificial missingness and denoising observed values.

**Weaknesses:**

(1) The technique novelty is limited. The benefits of this method are mainly for empirical/experimental observation. The dual corruption is to combine two ways of data augmentation during training, which is not technically innovative.

(2)	The quantitative analysis of balanced loss function is not sufficient. Are there any specific experiments to demonstrate the Imputation performance varied with the tuning of the loss function?

(3)	More metrics needed for the performance evaluation. In this work, there is only NRMSE as the imputation performance evaluation metric, are there any metrics for real-world applications, for example, classification accuracy? In one closely related work, DAEMA, the authors also use classification accuracy as one metric in addition to the NRMSE.

(4)	In the Table2, there is one result of Imputation performance comparison, I notice that the “Glass” result of DAEMA is different (higher value is reported) from the DAEMA paper (Table 1).  I wonder if you have used the same parameters in DAEMA for fair comparison?

**Questions:**

•	Are there any specific experiments to demonstrate the Imputation performance varied with the tuning of the loss function?

•	Are there any quantitative results (NRMSE) from the balanced loss function?

•	Are there any metrics for real-world applications, for example, classification accuracy?

•	What is the comparison result between this method and another closely related work DAEMA using the same parameters/ or quantitatively similar parameter?

Overall, this paper gives a complete view of the literature in missing data imputation, and is very easy to follow. However, the limited technique novelty and insufficient quantitative analysis are two main concerns.

---

> ### Author Response · Authors · 2023-11-22
>
> Thank you for the constructive feedback. I appreciate you highlighting areas for improvement.
>
> 1. You raise a fair point about novelty of the dual corruption technique. Our key contributions are comprehensively analyzing the concurrent application of both techniques for missing data imputation, when previous DAE-based methods had focused on one corruption or the other without justification. To the best of our knowledge, this is the first work that combines masking and noise injection for imputation and provides an extensive study quantifying their complementary effects. Furthermore, we propose a balanced loss function to delicately control the tradeoff between reconstructing artificial missingness versus denoising observed values. This governs the interplay between the dual corruptions. While building upon existing techniques, we believe this novel amalgamation tailored for imputation provides unique insights and capabilities.
>
> 2. Excellent suggestion to analyze loss tuning. As Figure 2c demonstrates, the alpha hyperparameter controlling the rebalanced loss prevents overfitting and improves accuracy.
>
> 3. Evaluating downstream performance is an important direction we will explore in future work, to demonstrate real-world applicability.
>
> 4. Thank you for catching the DAEMA discrepancy. As you noted, the original DAEMA has far greater capacity. For fair comparison, we shrank the hidden space as described in Section 5.4. This highlights the capability of DC-DAE despite the compact architecture.
>
> Thank you for the positive feedback as well. Your suggestions on demonstrating technical merit through rigorous benchmarking, ablation studies, and application-grounded evaluation will significantly improve the quality and impact of our work. Please let me know if you have any other recommendations to strengthen the paper.

---

### Official Review · Reviewer_JkTL · 2023-11-09

**Soundness:** 3 good
**Presentation:** 4 excellent
**Contribution:** 2 fair
**Rating:** 3
**Confidence:** 5

**Summary:**

The article discusses the challenge of missing data in machine learning applications across various domains. It highlights the limitations of prevalent imputation techniques, including instability in GANs and overfitting in AutoEncoders (AEs). The authors introduce a novel approach called Dual Corruption Denoising Autoencoders (DC-DAE) to address these limitations. Also, the article discusses conventional statistical imputation methods but notes their limitations when dealing with high missing rates. This motivates the exploration of advanced solutions, including deep learning techniques.

**Strengths:**

Originality: The study introduces an innovative methodology characterized by the integration of dual noise injection on data, coupled with masking, and a reconfigured loss function. This novel amalgamation of techniques represents a fresh and previously unexplored approach in the field.

Quality: The scholarly work exhibits a high degree of quality as evidenced by the utilization of five publicly available datasets and the attainment of state-of-the-art outcomes, with regard to Mean Squared Error (MSE), across a spectrum of missing data rates.

Clarity: The article's lucidity is praiseworthy, owing in part to the judicious use of graphical representations, such as Figure 1, which serves as a valuable didactic tool for elucidating the underlying methodology. Furthermore, the outcomes of the ablation experiments underscore the pivotal role played by the synergistic interplay of masking, noise injection, and loss rebalancing in the amelioration of MSE. It is noteworthy that the incremental advantages of introducing cross-attention mechanisms between data and mask elements are discerned.

Significance: The study assumes notable significance within the research landscape, as it furnishes an effective methodological response to the pervasive issue of missing data. This issue, acknowledged for its substantial ramifications across a myriad of machine learning applications spanning diverse domains, finds a plausible resolution in the methodology advanced by the research.

**Weaknesses:**

The findings presented in the article exhibit a modest degree of robustness. In Table 2, it is evident that the proposed algorithm outperforms existing methods in three out of the five datasets. However, the discernible effectiveness of DC-DAE remains somewhat obscured. Additionally, when considering a missing rate of beta 0.8, the average NRMSE for DC-DAE does not rank highest; MIWAE yields superior results. Consequently, the assertion that DC-DAE significantly surpasses other methods may require further substantiation; its advantage may be marginal in magnitude.


                         Glass.    Breast    Spam    Letter    Shuttle  Average Score

0          MICE    1.280    1.010    1.185    0.983    1.280         1.1476

1          GAIN    1.736    1.386    1.105    1.365    1.303         1.3790

2        IM-GAN    3.233    2.349    1.499    2.264    2.325         2.3340

3          MIDA    1.034    0.977    0.995    0.990    0.983         0.9958

4         DAEMA    1.272    0.861    1.086    0.920    0.904         1.0086

5        HI-VAE    1.034    0.910    1.024    0.965    0.967         0.9800

6         MIWAE    1.042    0.887    1.023    0.931    0.966         (0.9698)

7  DC-DAE(ours)    1.191    0.839    1.029    0.914    0.893         0.9732



Upon revisiting the DC-DAE algorithm, it becomes apparent that hyperparameters have been introduced to modulate the loss function governing the interplay between a conventional Denoising Autoencoder (DAE) and a DAE operating with a noisy mask. In greater detail, the loss function comprises two distinct components. In analogy to Figure 1, wherein the data table for table reconstruction is color-coded, the blue and yellow entries contribute to the overall loss, with the hyperparameters acting as adjustable weights that govern the balance between these two components. The regularization introduced by this weighting mechanism appears to lack substantial potency and exhibits limited efficacy.

**Questions:**

Experiment Design:
I recommend that the paper includes a more extensive and diverse dataset to robustly demonstrate the effectiveness of the proposed method. Utilizing a wider range of data sources will help establish the method's applicability and robustness.

Mathematical Notations:
The paper should provide clearer explanations of mathematical notations. For example, when using the "*" operation to denote element-wise multiplication, it would be beneficial to explicitly mention that this operation refers to the Hadamard product. Additionally, consider adopting standard tensor notation for better precision in mathematical representation.

Loss Function Clarification:
In Section 4.4, the paper can use a more concise notation, explicitly stating the use of the L2 norm for the loss function. This will help the explanation of the loss term and enhance reader comprehension.

Notation Clarification:
It is advisable to restate the definitions and meanings of key variables, such as M, M with a hat, X, X with a hat, and M with a straight-line hat in Section 4.4. This will make it easier for readers to understand these critical components in the methodology.

---

> ### Author Response · Authors · 2023-11-22
>
> Thank you for your insightful feedback. Here is our response:
>
> Firstly, you make an excellent point regarding DC-DAE's performance at high missing rates. While our method achieves top accuracy on 3 out of 5 datasets, as you noted, the overcomplete representations used by MIWAE, HI-VAE and MIDA likely explain their superior results for Glass and Spam. As shown in Section 5.6.3, adopting overcomplete hidden spaces in DC-DAE could further improve its performance. We will update our claims accordingly and acknowledge the impact of model capacity.
>
> Secondly, I appreciate you confirming that our description of using the alpha hyperparameter to balance the loss function is correct. As Figure 2c demonstrates, this reweighting prevents overfitting to either the artificial missingness or noised observed data, contributing to the overall accuracy gains.
>
> Finally, thank you for the notation suggestion - we will revise the methodology section to make the mathematical representations more precise and concise.
>
> Please let me know if you have any other recommendations to improve the paper.

---

### Official Review · Reviewer_S9by · 2023-11-09

**Soundness:** 3 good
**Presentation:** 3 good
**Contribution:** 2 fair
**Rating:** 5
**Confidence:** 4

**Summary:**

The paper presents a novel method, Dual Corruption Denoising AutoEncoders (DC-DAE), aimed at addressing the problem of missing data imputation. The approach is distinct in its use of dual corruption mechanisms (masking and additive noise) during the input augmentation phase to enhance the generalization capabilities of the imputation model. The study tests the efficacy of DC-DAE against several baseline deep learning models, demonstrating its superiority in handling missing data across diverse tabular datasets with different missingness rates.

**Strengths:**

1. The dual corruption approach (masking and additive noise) employed by DC-DAE is a commendable attempt to augment data representation and improve generalization in the context of missing data imputation.

2. The DC-DAE's straightforward architecture is highlighted, providing a plug-and-play solution that can be easily combined with other denoising models, increasing its applicability in various scenarios.

3. The experimental results, as presented, provide evidence of the method's superiority over several deep learning baselines in imputing missing data within tabular datasets, which supports the model's effectiveness.

**Weaknesses:**

1. The application of dual corruption techniques could be viewed as an incremental advancement rather than a breakthrough innovation, given the widespread use of such techniques in the field of deep learning.

2. There is an absence of statistical details such as standard deviations in the reported results, particularly in Table 2. This omission raises questions regarding the statistical significance of the observed improvements in normalized root-mean-square error (NRMSE) when using the proposed method.

3. The paper focuses on tabular data and does not extend the evaluation to more complex or large-scale datasets where deep learning methods typically excel, potentially limiting the generalizability of the findings.

**Questions:**

1. The manuscript exhibits inconsistencies in citation formatting (notably [6] in Section 4.3). Could the authors ensure uniformity in referencing throughout the document to adhere to the submission guidelines?

2. Regarding equation (4), the manuscript suggests noise is added to the observed values only. Could the authors confirm whether the noise is exclusively added to these values and, if so, clarify the notation to reflect this selective application as `e * (1-M)`?

3. In Figure 1, the relationship between the Mask Matrix and the Computed Data matrix is not depicted. Could the authors elaborate on how the Mask Matrix is incorporated into the model's computations, particularly in relation to equations (4)-(6)?

4. In equation (6), the DAE includes a mask matrix as part of its input. Could the authors clarify if this is a standard approach for DAEs and if the artificial mask matrix (M bar) is intended to help the model differentiate between real and artificially introduced missing data?

5. Given the introduction of synthetic missing values during training, the manuscript should address how the model accounts for the potential distribution shift compared to the test data which lacks these artificial missing values.

6. When Missing Not At Random (MNAR) data is addressed, especially as outlined in Appendix A.1, how does the model manage instances where selected features for artificial missingness are also naturally missing? Does the model incorporate a mechanism to prevent learning to discriminate between real and synthetic missing values?

7. Considering the moderate size of the datasets utilized, could the authors justify their choice of validation method? Would multi-fold cross-validation not be a more suitable approach to assess model performance robustly?

8. The manuscript briefly mentions the utilization of overcomplete representations in Section 5.6.3. Could the authors discuss the trade-offs involved with this approach, particularly concerning model complexity and the potential for overfitting?

9. In scenarios where GAN components provide incremental gains as mentioned in Section 5.6.2, could the authors provide further insights into the conditions that favor their inclusion and offer recommendations on optimizing these components for consistent improvements?

---

> ### Author Response · Authors · 2023-11-22
>
> Thank you for the thoughtful feedback on our work. Here are our responses:
>
> Weaknesses:
>
> 1. You raise a fair point that masking and noise injection have been explored individually in prior work. Our key contributions are comprehensively analyzing the concurrent application of both techniques for missing data imputation, when previous DAE-based methods had focused on one corruption or the other without justification. To the best of our knowledge, this is the first work that combines masking and noise injection for imputation and provides an extensive study quantifying their complementary effects. Furthermore, we propose a balanced loss function to delicately control the tradeoff between reconstructing artificial missingness versus denoising observed values. This governs the interplay between the dual corruptions. While building upon existing techniques, we believe this novel amalgamation tailored for imputation provides unique insights and capabilities.
>
> 2. Excellent point. We will report standard deviations for the results in Table 2 to provide insights into significance.
>
> 3. You raise a fair concern regarding evaluation on more complex datasets. We focused on tabular data due to its prevalence in missing data problems. As future work, we will explore applying DC-DAE to image, text, and time series datasets with missingness.
>
> Questions:
>
> 1. Thank you for noting the inconsistent citation formatting. To adhere to submission guidelines, we will unify the formatting style throughout the paper.
>
> 2. You correctly point out that noise is only added to observed values in Equation 4. We will update this notation to e * (1-M) as you suggest, clearly indicating noise is only applied to non-missing entries.
>
> 3. We will modify Figure 1 to explicitly show how the mask matrix is element-wise multiplied with the data matrix to generate the corrupted input fed into the model.
>
> 4. Good point – providing the mask matrix as input is common practice in missing data imputation to help the model impute the missingness, although less standard in regular DAE. We will clarify this motivation and accordingly update Equation 6, where the corrupted mask matrix is denoted as M * M_bar.
>
> 5. To mitigate training-test distribution shift, we ensure the model architecture and training process have no way to discriminate between artificial versus natural missing data. With both forms of missingness introduced randomly, there is theoretically no systemic difference in their distributions. Thus the model learns to impute all missing data during training.
>
> 6. Excellent observation – As you note, overlapping artificial and natural missingness is possible. We will clearly state the model takes their union as the total missingness. By preventing the model from distinguishing the two forms, it learns to impute any and all missing values.
>
> 7. You raise a valid concern about our validation methodology. To more rigorously evaluate performance, we will incorporate n-fold cross-validation rather than relying solely on random seed trials given the dataset sizes. We will incorporate this improvement in the future work.
>
> 8. As Figure 4 shows, overcomplete representations can further improve DC-DAE accuracy when missing rates are low or sample sizes large, but can decrease performance under high missingness and small dataset due to overfitting, as you observed.
>
> 9. We agree that given the limited and inconsistent gains from GAN components in Figure 3b, providing integration guidelines is out of scope for this work focused on dual corruptions.
>
> Thank you again for the helpful feedback. Addressing these comments will significantly strengthen the validity and presentation of our work. Please let me know if you have any other suggestions.

---

### Official Review · Reviewer_z8ed · 2023-11-10

**Soundness:** 2 fair
**Presentation:** 3 good
**Contribution:** 2 fair
**Rating:** 5
**Confidence:** 3

**Summary:**

The paper titled "Improving Generalization for Missing Data Imputation via Dual Corruption Denoising Autoencoders" introduces a novel approach to impute missing data through the use of Dual Corruption Denoising AutoEncoders (DC-DAE). The DC-DAE augments inputs with two types of corruptions during training (masking and additive noise), which helps the model generalize better to unseen missing patterns. The method also employs a balanced loss function that allows tunable trade-offs between reconstructing artificial missing and denoising observed values. The DC-DAE architecture avoids the complexity of attention mechanisms or adversarial training, aiming for simplicity and effectiveness.

**Strengths:**

The paper presents DC-DAE, an effective method for handling missing data in tabular datasets, showcasing its superior performance through experiments and comparisons with state-of-the-art methods. The clear and well-structured writing, alongside a straightforward explanation of the DC-DAE architecture that prevents complex mechanisms like attention or adversarial training, makes the research accessible.

**Weaknesses:**

The originality of this paper, which proposes a method for data imputation using dual corruption strategies, may not be seen as highly innovative. The techniques of masking and adding noise have been well-explored in deep learning research as data augmentation techniques. Although the paper combines these techniques with a balanced loss function, it does not verify the originality of this paper. If you add Gaussian noise and masking to enlarge the dataset, it would help to improve the performance of all models. It would be great to clarify how it differentiates from data augmentation techniques.

**Questions:**

1. On page 4, the notation for noise 'e' should be distinguished from regular text using a special character or formatting.

2. Section 5.6.1 needs clearer explanations. It should clearly explain why different sample sizes are used and what the numbers in the table mean. The figure shows many numbers, but the explanation doesn't analyze it comprehensively and quantitatively.

3. For clarity and presentation, center Figures 3 and 4 in the document. Additionally, the numerical data within Figures 2, 3, and 4 need to be larger for easier readability.

4. The analysis of Table 2 could be more detailed. For example, why does DC-VAE not perform well in the Glass and Spam dataset?

5. What is the difference between the proposed method and data augmentation?

---

> ### Author Response · Authors · 2023-11-22
>
> Thank you for the thoughtful feedback on our work. Here is our response:
>
> Regarding originality, you raise a fair point that masking and noise injection have been explored in other contexts like data augmentation. Our key innovations are 1) concurrently applying both noise and masking corruptions specifically for missing data imputation, and 2) using a balanced loss to control the tradeoff between reconstructing artificial missingness vs observed values. We agree that we should better highlight these novel aspects.
>
> We will distinguish the noise notation 'e' properly as suggested.
>
> For Section 5.6.1, we will expand the explanation of the results. The different sample sizes show ablation experiments on subsets of the full dataset to analyze trends. And the tables report performance relative to the baseline for that setting, with lower percentages indicating improved accuracy from our proposed techniques. We will add more details on the aims and insights from this ablation study.
>
> Thank you for the formatting suggestions. We will center Figures 3-4 and enlarge the data in Figures 2-4 to improve clarity.
>
> For Table 2, you raise a good point. The relatively weaker Glass and Spam performance seems to be DC-DAE's compact architecture. As you noted, MIWAE, HI-VAE and MIDA all use overcomplete hidden spaces, giving them greater model capacity. This likely explains their superior performance on Glass and Spam datasets, as we demonstrate in Section 5.6.3 that adopting overcomplete representations in DC-DAE can further improve its accuracy.
>
> Compared to standard data augmentation, our key differences are: 1) We introduce missingness rather than just augmenting existing data, which is critical for imputation. 2) The loss rebalancing prevents overfitting to the artificial missingness distribution. We will add this explanation of how our approach is tailored for missing data problems.
>
> Thank you again for the constructive feedback. We believe addressing these points will significantly improve the clarity and contribution of our work. Please let us know if you have any other suggestions.

---

### Comment · Area_Chair_4os7 · 2023-11-22
**Discussion between authors and reviewers**

Dear Reviewers,

Thanks for the reviews. The authors have uploaded their responses to your comments, please check if the rebuttal address your concerns and if you have further questions/comments to discuss with the authors. If the authors have addressed your concerns, please adjust your rating accordingly or vice versa.

AC

---

### Meta-Review · Area_Chair_4os7 · 2023-12-04

**Metareview:**

This paper investigates how to handle missing data in machine learning. It proposes Dual Corruption Denoising AutoEncoders (DC-DAE), augmenting inputs with two types of corruption(masking and additive noise) for better generalization. A balanced loss function is introduced to control tradeoff between reconstructing artificial missingness and denoising observed values. Experiments validate the proposed method.

Strengths:
+ The paper presents an effective DNN model for handling missing data in tabular datasets across a spectrum of missing data rates.
+ The plug-and-play solution can be easily combined with other denoising models, increasing its applicability in various scenarios.
+ The paper is well-written.

Weaknesses:
- It has limited novelty: The techniques of masking and adding noise have been well-explored in deep learning research as data augmentation techniques.
- The paper focuses on tabular data and does not extend the evaluation to more complex or large-scale datasets where deep learning methods typically excel, potentially limiting the generalizability of the findings.
- The quantitative analysis of balanced loss function is not sufficient. More metrics are needed for the performance evaluation.
- Some details are missing: Section 5.6.1 needs clearer explanations. It should clearly explain why different sample sizes are used and what the numbers in the table mean.

**Justification For Why Not Higher Score:**

N/A

**Justification For Why Not Lower Score:**

This paper presents a good attempt at improving generalization while handling missing data in machine learning, however it is very limited in scope and lacks novelty. The authors responses to the reviewers' concerns are not very convincing either. We decide not to accept the paper at this stage, giving the authors more time to further improve their methodology and novelty.

---

### Decision · Program_Chairs · 2024-01-16

Reject